# A Bayesian Nonparametric View
# on Count-Min Sketch

**Diana Cai**
Princeton University
dcai@cs.princeton.edu

**Michael Mitzenmacher**
Harvard University
michaelm@eecs.harvard.edu

**Ryan P. Adams**
Princeton University
rpa@princeton.edu

## Abstract

The count-min sketch is a time- and memory-efficient randomized data structure that provides a point estimate of the number of times an item has appeared in a data stream. The count-min sketch and related hash-based data structures are ubiquitous in systems that must track frequencies of data such as URLs, IP addresses, and language $n$-grams. We present a Bayesian view on the count-min sketch, using the same data structure, but providing a posterior distribution over the frequencies that characterizes the uncertainty arising from the hash-based approximation. In particular, we take a nonparametric approach and consider tokens generated from a Dirichlet process (DP) random measure, which allows for an unbounded number of unique tokens. Using properties of the DP, we show that it is possible to straightforwardly compute posterior marginals of the unknown true counts and that the modes of these marginals recover the count-min sketch estimator, inheriting the associated probabilistic guarantees. Using simulated data and text data, we investigate the properties of these estimators. Lastly, we also study a modified problem in which the observation stream consists of collections of tokens (i.e., documents) arising from a random measure drawn from a stable beta process, which allows for power law scaling behavior in the number of unique tokens.

## 1 Introduction

Modern software systems often involve large data streams [20] such as text queries, real-time network traffic, financial data, and social media activity. These systems are often required to detect anomalous data or report the frequencies of events and patterns in the stream. When processing these high-volume data streams, it is critical to compactly represent the data so that these analyses can be efficiently extracted. Ideally, it would be possible to estimate useful properties of the data stream in only a single pass using an amount of memory that is of constant size.

These practical desiderata for large-scale, real-time data analysis have led to the idea of constructing a *sketch*: a randomized data structure that can be easily updated and queried for approximate statistics of the stream. Variants of sketching ideas have found applications in many areas, including machine learning [1], security [10], and natural language processing [15]. Of particular interest has been the problem of estimating the frequency of tokens in a data stream (e.g., Misra and Gries [19], Charikar et al. [4], Cohen and Matias [6], Cormode and Muthukrishnan [8]), and a notable approach to this problem is the *count-min* sketch [8] (and its cousins such as the counting Bloom filter [13]), which uses random hash families to approximate these counts.

The count-min sketch is appealing because it successfully achieves the goal of using a compressed representation to save space in storing approximate frequency statistics of the data stream, with provable performance guarantees on the answers returned by those queries. Nevertheless, there are several aspects of the count-min sketch that might be improved upon by taking a different probabilistic view. First, the count-min sketch provides only point estimates of the statistics of interest, even

though the hashing procedure may induce substantial uncertainty. This uncertainty is particularly salient when estimating infrequent events. Second, the guarantees associated with hash-based data structures typically assume a finite universe of possible tokens in the stream. We would expect real data streams to have an unbounded number of unique tokens, as, e.g., people invent new hashtags and construct new web pages. Third, we often have *a priori* knowledge of the statistics of the data stream, and it is desirable to incorporate this knowledge into the estimates.

Here we instead take a Bayesian nonparametric view on the estimates arising from the count-min sketch data structure. We assume that the tokens in the stream are drawn from an unknown discrete distribution, and that this distribution has a Dirichlet process prior. The unique projective properties of the Dirichlet process interact elegantly with the partitioning induced by the random hashes. This, combined with the simple form of the resulting predictive distribution, makes it possible to reason straightforwardly about the posterior over the unknown true number of counts of a token, given the counts stored in the data structure. Notably, the maximum *a posteriori* estimate arising from this procedure recovers the count-min sketch estimator for some regimes of the Dirichlet process prior, and other posterior-derived point estimates can be viewed as "count-min sketch with shrinkage." The Bayesian nonparametric view also leads to useful alternative data structures with strong similarities to count-min sketch; we examine one such example in which the stream is composed of "documents" from a random measure induced by a stable beta-Bernoulli process.

The paper is structured as follows. We first review count-min sketch and the resulting frequentist guarantees on the point estimates. Next, we review the Dirichlet process, revisiting the assumptions about the sketched data stream, leading to a form for the posterior marginals over the counts. We then examine the properties of these posterior marginals, relating them to the classical count-min sketch estimators in theory and simulation. Finally, we propose an alternative nonparametric approach based on the stable beta process, which enables the modeling of power law behavior in the stream.

## 2 The Count-Min Sketch

The count-min (CM) sketch [8] is a randomized data structure that uses random hashing to approximate count statistics of a data stream of tokens. Let $x_1, x_2, \ldots, x_M$ be an arbitrary stream of tokens taking values in a set $\mathcal{V}$, e.g., language $n$-grams, IP addresses, hashtags, or URLs. A *point query* estimates $\eta_v$, the number of times the token of type $v \in \mathcal{V}$ has appeared in the stream. The goal of a sketch is to estimate such quantities without explicitly storing the elements or the counts.

In the classical count-min sketch, $N$ hash functions $h_n : \mathcal{V} \to [J]$, where $[J] := \{1, \ldots, J\}$ and $|\mathcal{V}| \gg J$, are chosen uniformly at random from a *pairwise independent* hash family $\mathcal{H}$. That is, a random $h \in \mathcal{H}$ has the property that for all $v_1, v_2 \in \mathcal{V}$, such that $v_1 \neq v_2$, the probability that $v_1$ and $v_2$ hash to values $j_1, j_2 \in [J]$ is

$$\Pr_{h \in \mathcal{H}} (h(v_1) = j_1, h(v_2) = j_2) = \frac{1}{J^2}.$$

The sketch data structure $C = [c_{n,j}]_{n \in [N], j \in [J]}$ is an array of counts of size $N \times J$. When an observation of type $v$ arrives, the sketch is updated: for all $n \in [N]$, the counter associated with $v$ via $h_n$ is incremented via $c_{n,h_n(v)} \leftarrow c_{n,h_n(v)} + 1$. Then the point query of a new token $x$ returns the minimum count over all of the hash functions: $\hat{\eta}_x^{\mathrm{CM}} = \min_{n \in [N]} c_{n,h_n(x)}$. That is, it returns the count associated with the fewest collisions. This provides an upper bound on the true count. For an arbitrary data stream with $M$ tokens, the CM sketch satisfies the following probabilistic guarantee.

**Theorem 1** (Cormode and Muthukrishnan [8], Theorem 1). *Let $J = \lceil \frac{e}{\epsilon} \rceil$ and $N = \lceil \log \frac{1}{\delta} \rceil$, with $\epsilon > 0, \delta > 0$. Then the estimate of the count $\hat{\eta}_x^{\mathrm{CM}}$ satisfies $\hat{\eta}_x^{\mathrm{CM}} \geq \eta_x$ and with probability at least $1 - \delta$, the estimate satisfies $\hat{\eta}_x^{\mathrm{CM}} \leq \eta_x + \epsilon M$.*

The count-min sketch can also be used to compute other queries, such range and inner product queries; though we focus on the point query in this work, analogous Bayesian reasoning can also be applied to these other types of queries. Count-min sketches have been considered in the context of specific token distributions but without Bayesian reasoning; for example, one can derive better probabilistic bounds when the distribution is known to come from a "power law" distribution [7]. An extension of the classical CM sketch that works well in practice is conservative updates [11]; in our setting, we observe one token at a time, and so the conservative update is equivalent to simply incrementing the minimum counter(s). For more background on the count-min sketch, extensions, and variants, see Cormode et al. [9, Sec. 5] and references therein.

# 3 Bayesian Estimates from the Count-Min Sketch

The count-min sketch offers a remarkable balance between efficiency and accuracy, but it nevertheless achieves its computational performance by throwing information away. This loss of information occurs via hash collisions. Although we do not know when these collisions have occurred and damaged our estimates, it is possible to reason about the uncertainty arising from the possibility of collision. Taking a Bayesian view, for a given token $x$, the values in the data structure $\{c_{n,h_n(x)}\}_{n=1}^N$ are the observations, and we wish to induce a posterior distribution over the true count $\eta_x$ by conditioning upon them. To compute this posterior distribution, we must identify a prior over the counts, and we must also reason about the count-min sketch as a likelihood function. In this section, we describe how to compute the posterior count under a Dirichlet process (DP) prior, marginalizing out the random measure. We do not alter the underlying count-min sketch data structure. The proofs for this section are presented in Appendix D and Appendix E.

## 3.1 The Dirichlet Process

As before, we assume the tokens are from a set $\mathcal{V}$. Let $G$ be a continuous probability base measure on the measurable space $(\mathcal{V}, \mathcal{F})$, and let $\alpha > 0$ be the *concentration parameter*. The base measure is the mean of the prior on the unknown token generating distribution, and $\alpha$ can be thought of as an "inverse variance" that determines how similar the random measures are to $G$. The Dirichlet process [14] is defined by the property that for all finite partitions of $\mathcal{V}$, the distribution over the measures of those partitions is Dirichlet. That is, for all finite, measurable partitions $A_1, \ldots, A_N \subset \mathcal{V}$, the random measure $H \sim \mathrm{DP}(G, \alpha)$ has the property

$$H(A_1), \ldots, H(A_N) \,|\, G, \alpha, \{A_n\}_{n=1}^N \sim \mathrm{Dirichlet}(\alpha G(A_1), \ldots, \alpha G(A_N)). \qquad (1)$$

This implies for the present construction that renormalized restrictions of the random measure are also Dirichlet process distributed. That is, if $A \subset \mathcal{V}$ and $H_A$ is the marginal renormalized random measure on $A$ induced by the DP on $\mathcal{V}$ with concentration parameter $\alpha$ and base measure $G$, then $H_A \sim \mathrm{DP}(G_A/G(A), \alpha G(A))$, where $G_A$ is the base measure restricted to $A$. This property can also be derived by observing that the Dirichlet process is a normalized gamma process, which has a Poisson process representation admitting the Poisson coloring theorem [17] (see Appendix C).

The Dirichlet process produces random measures that are discrete with probability one. This discreteness implies that observations drawn from a DP random measure have positive probability of having repeated values. If we take data with the same value to be part of the same group, then this provides a random partition of $M$ data points. Integrating out the random measure $H$ results in an object called the *Chinese restaurant process* (CRP) [2], which induces an exchangeable random partition of a finite data set. The CRP is most commonly discussed via its predictive distribution, which we describe in the language of our token stream: if $x_m$ is the $m$th token in the stream, $\eta_v$ is the number of previous tokens taking that value, then $x_m$ is distributed according to

$$\Pr(x_m = v \,|\, x_1, x_2, \ldots, x_{m-1}, \alpha) = \begin{cases} \frac{\eta_v}{\alpha + m - 1} & \text{if } v \text{ is a previously seen token} \\ \frac{\alpha}{\alpha + m - 1} & \text{if } v \text{ is a novel token} \end{cases}. \qquad (2)$$

## 3.2 Bayesian Point Query: a Distribution Over the Count of a Token

In the classical CM sketch setting, the point query for a token returns the minimum value in the associated counters. In a Bayesian setting, the point query induces instead a posterior distribution over the unknown true counts of a token, conditioned on the observed counts in the counter array $C$. Returning a distribution over possible counts allows us to quantify the uncertainty in the count of a token. While our goal is not to compute a point estimate for the count, such an estimate can be obtained from the posterior count distribution by considering, for instance, the posterior mean, median, or mode (MAP), the properties of which we describe in Section 3.4.

The posterior of the count summarized in Theorem 2 relies on two results: (1) the distribution of tokens in each bucket is a Chinese restaurant process with parameter $\alpha/J$, and (2) the posterior of a single hash can be obtained from the $\mathrm{CRP}(\alpha/J)$ distribution (Proposition 1). A key assumption informing these results is that the point queries are drawn from the same distribution as the stream,

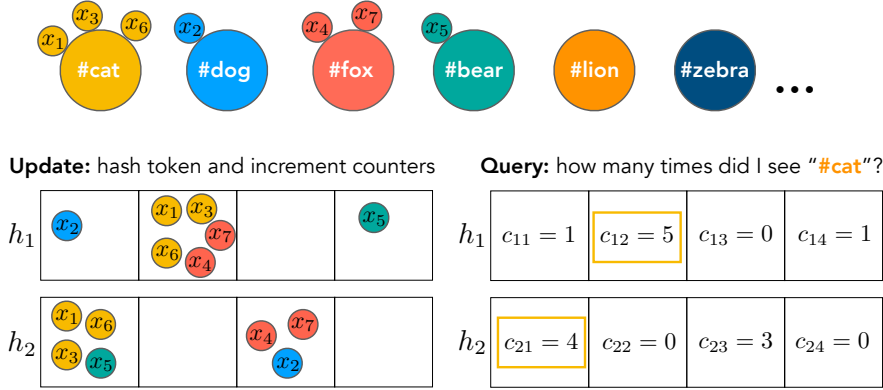

Figure 1: **Top:** Tokens are generated from a $\mathrm{CRP}(\alpha)$ distribution, where the large circles represent tables from the Chinese Restaurant analogy, each labeled by the token type, and small circles denote the tokens in the data stream $x_1, x_2, \ldots$. **Bottom left:** The update operation hashes tokens and increments the associated counters, and information regarding the type of the token (denoted by the colored circles) is therefore lost when making a query. Each bucket in the hash now follows a $\mathrm{CRP}(\alpha/J)$ distribution. **Bottom right:** The Bayesian point query for each token uses the respective counters (denoted by the colored rectangles) as observations for the posterior count distribution.

i.e., we make queries about tokens as they come in on the stream. This assumption makes it possible to reason about the queries via the predictive distribution induced by the Chinese restaurant process.

The count-min sketch data structure can be thought of as creating $N$ collections of $J$ "buckets," each of which aggregates the counts for all $x$ where $h_n(x) = j$. The posterior distribution of interest essentially tries to undo the collisions that caused this aggregation. Assume that $\mathcal{H}$ is a *truly random hash family*, i.e., for $h : \mathcal{V} \to [J]$ drawn uniformly at random from $\mathcal{H}$, the random variables $(h(x))_{x \in \mathcal{V}}$ are i.i.d. uniform over $[J]$. Note that the count-min sketch bounds from Theorem 1 only depend on pairwise-independent hash families. We discuss this further in Section 3.5.

Uniformity of $h \sim \mathcal{H}$ implies that each $h_n$ induces a $J$-partition of $\mathcal{V}$, and the measure with respect to $H$ of each class of the $J$-partition is $1/J$. Thus, using the restriction property of the Dirichlet process, the hash function turns a *global* DP governing the distribution over tokens in the stream into a collection of $J$ *bucket-specific* DPs that govern only the tokens that hashed there. Moreover, within each of these buckets, the unknown random measure can be marginalized out, and the structure can be manipulated as the simpler Chinese restaurant process. The CRP has precisely the structure that we seek to reason about in our point query: some finite number of objects have hashed into this bucket, and we need to construct a posterior on how those might be partitioned into groups of identical tokens. In the parlance of the Chinese restaurant metaphor, we know how many people have come into the restaurant, but we have lost the information about who is sitting at which table; a new customer comes in (the query) and wants to know how many people should be sitting at their table given the total number of customers.

Let $\alpha$ be the global Dirichlet process concentration parameter. Then, the bucket-specific DP parameter is $\alpha' := \alpha/J$. The posterior distribution of interest is

$$\Pr(\eta_x = k \mid \{c_{n,h_n(x)}\}_{n=1}^N, \alpha) \propto \Pr(\eta_x = k \mid \alpha) \prod_{n=1}^N \Pr(c_{n,h_n(x)} \mid \eta_x = k, \alpha). \tag{3}$$

However, there is a simpler interpretation for each of the terms in the likelihood function. Consider the random partitioning of $c$ items according to a CRP. For a fixed such partition, the predictive distribution of Equation (2) induces a distribution over which subset a new item will join, and so the existing size of that subset is also a random variable. That existing size is precisely the quantity $\eta_x$ we seek to estimate. For a fixed (integer) partition $\pi$, let $\pi_k$ be the number of subsets of size $k$,

where $\sum_{k=1}^{c} k\pi_k = c$. The distribution over the size of the subset a new item joins is then

$$\Pr(\eta_x = k \mid \pi, c, \alpha) = \begin{cases} \frac{\alpha}{\alpha+c} & \text{if } k = 0 \quad \text{[creates new size-one subset]} \\ \pi_k \frac{k}{\alpha+c} & \text{if } k > 0 \quad \text{[}\pi_k \text{ opportunities for CRP predictive]} \end{cases}. \tag{4}$$

The additional complication however is that, unlike the typical CRP situation, the partitioning itself is unknown, and so we must marginalize over it under the prior when computing the distribution on $\eta_x$. However, we can recognize this as simply using the expected number of subsets of a particular size:

$$\Pr(\eta_x = k \mid c, \alpha) = \sum_{\pi} \Pr(\eta_x = k \mid \pi, c, \alpha) \Pr(\pi \mid c, \alpha) = \begin{cases} \frac{\alpha}{\alpha+c} & \text{if } k = 0 \\ \frac{1}{\alpha+c} \sum_{\pi} k\pi_k \Pr(\pi \mid c, \alpha) & \text{if } k > 0 \end{cases}$$

$$= \frac{1}{\alpha + c} \begin{cases} \alpha & \text{if } k = 0 \\ k\overline{\pi}_k & \text{if } k > 0, \end{cases} \qquad \text{where } \overline{\pi}_k := \mathbb{E}[\pi_k] = \sum_{\pi} \pi_k \Pr(\pi \mid c, \alpha).$$

The probability of the partition $\pi$ is the EPPF [21] multiplied by the unordered multinomial coefficient:

$$\Pr(\pi \mid c, \alpha) = \frac{c! \, \Gamma(\alpha)}{\Gamma(c+\alpha)} \prod_{k=1}^{c} \frac{\alpha^{\pi_k}}{k^{\pi_k} \pi_k!} = \frac{c! \, \alpha^{\sum_{k=1}^{c} \pi_k} \Gamma(\alpha)}{\Gamma(c+\alpha)} \prod_{k=1}^{c} \frac{1}{k^{\pi_k} \pi_k!}, \tag{5}$$

where $\Gamma(\cdot)$ is the gamma function. For the Dirichlet process, this is also known as the Ewens's sampling formula. The following lemma gives the required expectation $\overline{\pi}_k$.

**Lemma 1.** *Let $\pi$ be a $\mathrm{CRP}(\alpha)$-distributed random partition of $c$ items and $\pi_k$ be the number of subsets in that partition of size $k$. The expected number of size-$k$ subsets is*

$$\overline{\pi}_k := \mathbb{E}[\pi_k] = \frac{\alpha}{k} \frac{\Gamma(c + \alpha - k)\Gamma(c+1)}{\Gamma(c+\alpha)\Gamma(c+1-k)}. \tag{6}$$

*Proof.* Applying Proposition B.1 to the rate measure $\nu(dp) = \alpha p^{-1}(1-p)^{\alpha-1} dp$, results in

$$\mathbb{E}[\pi_k] = \alpha \frac{\Gamma(c+1)}{\Gamma(c-k+1)\Gamma(k+1)} \int_0^1 p^{k-1}(1-p)^{c-k+\alpha-1} \, dp = \frac{\alpha}{k} \frac{\Gamma(c+1)\Gamma(c-k+\alpha)}{\Gamma(c-k+1)\Gamma(c+\alpha)}.$$

$\square$

This result matches the expectation given in Watterson [23, Eq. 2.22], which computes the expectation as a special case of the factorial moments for the Ewens's sampling process [12]. We can now reframe the derivation in terms of the partitions induced in a Dirichlet process by a hash function.

**Proposition 1.** *Let $h : \mathcal{V} \to [J]$ be a hash function drawn uniformly from a truly random hash family $\mathcal{H}$. Suppose that a stream of symbols is drawn from a random Dirichlet process measure with continuous base measure on $\mathcal{V}$ and concentration parameter $\alpha$. When the $M$th item $x_M$ arrives, let its hashed value be $h(x_M) = j$. Define $c_j$ to be the number of previous items in the stream that have also hashed to $j$, i.e., $c_j = \sum_{m=1}^{M-1} \mathbb{1}[h(x_m) = j]$. Given $c_j$, the posterior distribution over the true number of previous occurrences of items with the same type as $x_M$ is*

$$\Pr(\eta_{x_M} = k \mid c_j, \alpha) = \frac{1}{\alpha/J + c_j} \begin{cases} \frac{\alpha}{J} \frac{\Gamma(c_j+1)\Gamma(c_j-k+\alpha/J)}{\Gamma(c_j-k+1)\Gamma(c_j+\alpha/J)} & \text{if } k > 0 \\ \alpha/J & \text{if } k = 0 \end{cases}. \tag{7}$$

Due to the independence assumption on the hash family $\mathcal{H}$, the full posterior count is proportional to the product of the individual hash function posteriors from Proposition 1, as summarized below.

**Theorem 2.** *Let there be $N$ hash functions $h_1, \dots, h_N$ each drawn uniformly at random from a truly random hash family $\mathcal{H}$. Define a Dirichlet process stream of tokens as in Proposition 1. When the $M$th item $x_M$ arrives, let its $n$th hashed value be $h_n(x_M) = j_n$. Define $c_{n,j_n}$ to be the number of previous items in the stream that the $n$th hash has also hashed to $j_n$, i.e., $c_{n,j_n} = \sum_{m=1}^{M-1} \mathbb{1}[h_n(x_m) = j_n]$. Given the counts $c_{1,j_1}, \dots, c_{N,j_N}$, the posterior distribution over the true number of previous occurrences of items with the same type as $x_M$ is*

$$\Pr(\eta_{x_M} = k \mid c_{1,j_1}, \dots, c_{N,j_N}, \alpha) \propto \prod_{n=1}^{N} \frac{1}{\alpha/J + c_{n,j_n}} \begin{cases} k \, \overline{\pi}_{k,c_{n,j_n}} & \text{if } k > 0 \\ \alpha/J & \text{if } k = 0, \end{cases} \tag{8}$$

*where*

$$\overline{\pi}_{k,c_{n,j_n}} := \frac{\alpha}{Jk} \frac{\Gamma(c_{n,j_n} + 1)\Gamma(c_{n,j_n} - k + \alpha/J)}{\Gamma(c_{n,j_n} - k + 1)\Gamma(c_{n,j_n} + \alpha/J)}. \tag{9}$$

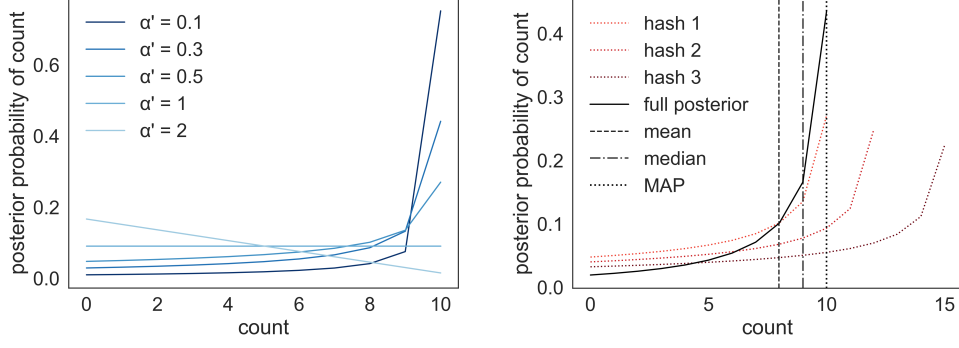

Figure 2: **Left:** The posterior probability of a single hash function conditioned on $c = 10$ and varying $\alpha'$. **Right:** The posterior of 3 hash functions with observed counts of 10, 12, 15, denoted by the red dotted curves, and with $\alpha' = 0.5$. The solid black curve denotes the normalized product of the individual hashes' posteriors. Vertical lines denote the posterior mean, median, and MAP.

## 3.3 Estimating $\alpha$ via Empirical Bayes

In practice, $\alpha$ is unknown and must be estimated from the data. Although information is lost due to hash collisions, it is nevertheless possible to construct a marginal likelihood that exactly accounts for this censoring: the finite projection property of the Dirichlet process specifies what the distribution should be of the hashed counts in each of the buckets. That is, the hash function forms a partition of $\mathcal{V}$, and Equation (1) specifies the resulting marginal Dirichlet distribution. For a single hash, we can integrate that finite-dimensional distribution against the multinomial counts $c_{n,1}, \ldots, c_{n,J}$, resulting in a Dirichlet-multinomial distribution with symmetric parameters $\alpha/J$. With a fully random hash family, $N$ hashes lead to a factorial marginal likelihood after seeing $M$ items in the stream:

$$\Pr(\{\{c_{n,j}\}_{j=1}^J\}_{n=1}^N \,|\, \alpha) = \prod_{n=1}^N \frac{M!\,\Gamma(\alpha)}{\Gamma(M+\alpha)} \prod_{j=1}^J \frac{\Gamma(c_{n,j}+\alpha/J)}{c_{n,j}!\,\Gamma(\alpha/J)} \,. \tag{10}$$

Maximizing the marginal likelihood can be done efficiently using a variety of techniques such as Newton-Raphson, as it is one-dimensional and log-concave in $\log(\alpha)$ [18].

## 3.4 Bayesian CM Sketch Point Estimates

Point estimates such as the mean $\hat{\eta}_x^{(\mathrm{mean})}$, median $\hat{\eta}_x^{(\mathrm{med})}$, and MAP $\hat{\eta}_x^{(\mathrm{MAP})}$ can be derived from the posterior $\Pr(\eta_k = k | C, \alpha)$ over the true item frequencies. The following lemma is useful for understanding the behavior of these point estimators and the effect of $\alpha$ relative to the size of the data structure as determined by $J$.

**Lemma 2.** *For $\alpha < J$, the function $\Pr(\eta_x = k | c_{n,j_n}, \alpha/J)$ is strictly increasing on $k = 0, \ldots, c_{n,j_n}$.*

Monotonicity of the posterior ensures the following key result that equates the classical CM sketch estimator with the MAP estimator.

**Proposition 2.** *For $\alpha < J$, we have $\hat{\eta}_x^{\mathrm{MAP}} = \hat{\eta}_x^{\mathrm{CM}}$, and $\hat{\eta}_x^{\mathrm{mean}} \leq \hat{\eta}_x^{\mathrm{med}} \leq \hat{\eta}_x^{\mathrm{MAP}}$.*

Crucially, this equality result provides an alternative view of the classical count-min sketch estimator as the maximum *a posteriori* estimate when $\alpha < J$. The ordering of point estimates also induces an interpretation on $\alpha$ as a shrinkage parameter for the count-min sketch.

In Figure 2, we plot the posterior as a function of the count to provide additional intuition about the structure of the estimate. The left plot shows the posterior varying $\alpha' = \alpha/J$. The right plot shows for a fixed $\alpha' = 0.5$, the individual hash posteriors and the full posterior for hypothetical observed counts of $10, 12$, and $15$. We can see that the posterior functions for $\alpha' < 1$ are monotonically increasing, and that the MAP estimator is equal to the minimum count, i.e., the CM sketch estimator.

### 3.5 Discussion

**Probabilistic interpretation of the CM Sketch.** In this paper we have reinterpreted a successful and widely-deployed randomized algorithm in a Bayesian probabilistic light. A particularly salient application of the Bayesian estimates of the CM sketch is scalable Bayesian inference via approximate sufficient statistics. In many Bayesian applications for discrete streaming data, the *sufficient statistics* for the likelihood are count statistics of the stream, which are too large to be stored in memory. One could directly approximate the sufficient statistics with, e.g., the CM point estimator, but this does not take into account the uncertainty of the count arising from hash collisions. Instead, using the posterior output of our method, one can build the uncertainty of the count into the model by integrating out the unobserved true count.

**Practical considerations for random hash families.** Thus far, we have used the mathematically convenient setting where all hash functions are perfectly random. In practice, real-world hash functions generally perform as if they were random. In a data stream of $M$ of tokens, consider the unique tokens $v_1, \ldots, v_k$ that appear in the stream. We would like the $h(v_i)$ to behave as though $h$ were a random function. The collection of $h(v_i)$ values will be $\epsilon$-close to uniform random values, even for hash functions $h$ chosen from a pairwise independent family (which are natural to use in practice), as long as the token types $v_i$ have sufficient entropy [5]. That is, we require much less randomness from the hash function, as long as the unique tokens have sufficient randomness. The relationship between $\epsilon$ and the entropy for pairwise and 4-wise independent hash functions are given in detail in Chung et al. [5]; here, the main point is that practical hash functions yield only small perturbations in our analysis from perfect hash functions.

**Parallel sketching algorithms.** Our method inherits the key property of a *linear sketch* from the CM sketch: that is, we can divide up the stream, compute a sketch posterior on each subset of the stream, and convolve the marginals to get back properties of the original stream. This leads to natural parallel algorithms for computing posterior estimates for large-scale streaming applications.

## 4 Sketching Beta-Bernoulli Process Counts

Although we have focused on the case where the stream consists of individual tokens, the CM sketch also allows for a vector of tokens to arrive at each step in the stream. We consider the case where at each step of the stream, a set of tokens arrives. It is natural to think of these sets as documents and the tokens as unique words, e.g., a stream of tweets each containing a small set of hashtags. The query of interest is then to determine how many documents a particular token appeared in. Bayesian nonparametric feature models, such as the Indian buffet (beta-Bernoulli) process, provide natural document-centered generalizations of the Dirichlet process sketch. Appendix F contains the detailed derivations for this section.

Suppose we have a model given by a stable beta-Bernoulli process random measure [22, 3] $B \sim \mathrm{BP}(\alpha, \gamma, d)$, where $\alpha > 0$ is the concentration parameter, $\gamma > 0$ is the mass parameter, and $d \in [0, 1)$ is the discount parameter satisfying $d > -\alpha$. At each step in the stream, a sparse binary vector $x_m \mid B \overset{\mathrm{iid}}{\sim} \mathrm{BeP}(B)$ arrives, where $x_m$ is an infinite-dimensional binary vector, with $x_{mi} = 1$ if the $i$th token is present in observation $x_m$ and 0 otherwise.

The data structure is essentially the same as the count-min sketch on the flattened stream. For each observation $x_m$, we hash the token and then increment the associated count that the token is hashed to. The goal of the estimator is, for a token appearing in a new observation $x_{M+1}$ from the stream, we want to return an estimate for how many documents that token has previously appeared in. Similar to the Dirichlet process, we can again treat the distribution of each hash function as its own beta-Bernoulli process, as a beta process can be represented by a Poisson point process, and therefore we can apply the Poisson coloring theorem to get $J$ independent beta-Bernoulli processes with mass parameter $\gamma' = \gamma/J$. Asymptotically, the number of unique tokens is $\frac{\gamma}{d} \frac{\Gamma(1+\alpha)}{\Gamma(\alpha+d)} M^d$ in the size of the stream $M$. One of the appealing properties of this construction is that this allows for power-law behavior in the number of unique tokens, as determined by the parameter $d$ [22]. Analogous to the Chinese restaurant process construction for the Dirichlet process in the partition case, the 3-parameter Indian buffet process (IBP) gives the probability of seeing an existing token $i$ in the next document as $\Pr(x_{M+1,i} = 1 | x_{1,i}, \ldots, x_{M,i}) = \frac{\eta_i - d}{M + \alpha}$, and then $\mathrm{Pois}(\gamma \frac{\Gamma(1+\alpha)\Gamma(M+\alpha+d)}{\Gamma(M+1+\alpha)\Gamma(\alpha+d)})$ novel tokens are drawn.

For a single hash $h$ and new observed token $v \in \mathcal{V}$, the conditional probability of the count $\eta_v$ being $k$ given both 1) the total the number of tokens that hashed into the same bucket $c_{h(v)}$, and 2) the number of tokens hashed into that bucket with count $k$ (as before, denoted $\pi_k$) is:

$$\Pr(\eta_v = k \mid \pi_k, c_{h(v)}) = \begin{cases} \pi_k \frac{k-d}{c_{h(v)}+\alpha} & \text{if } k > 0 \\ \frac{\Gamma(1+\alpha)\Gamma(c_{h(v)}+\alpha+d)}{\Gamma(c_{h(v)}+1+\alpha)\Gamma(\alpha+d)} & \text{if } k = 0 \end{cases}.$$

The individual hash posterior is therefore

$$\Pr(\eta_v = k \mid c_{h(v)}) = \begin{cases} \bar{\pi}_k \frac{k-d}{c_{h(v)}+\alpha} & \text{if } k > 0 \\ \frac{\Gamma(1+\alpha)\Gamma(c_{h(v)}+\alpha+d)}{\Gamma(c_{h(v)}+1+\alpha)\Gamma(\alpha+d)} & \text{if } k = 0 \end{cases}, \tag{11}$$

where $[\bar{\pi}_k = \mathbb{E}[\pi_k]]$ is the expected number of tokens that have been seen $k$ previous times under the IBP prior, *constrained to there being $c_{h(v)}$ total tokens*. In the Indian buffet process metaphor, this asks for the expected number of dishes that have been tasted $k$ times, given the total number of dish tastings. Unfortunately, this additional constraint makes this expectation challenging to compute. However, if we assume that the expected number of tokens per document is much smaller than $J$, then the probability of a single document having a collision between its tokens is small. We therefore make the following approximation for $c$ total tokens:

$$\bar{\pi}_k := \mathbb{E}[\bar{\pi}_k] \approx \gamma \frac{\Gamma(c+1)\Gamma(1+\alpha)\Gamma(k-d)\Gamma(c-k+\alpha+d)}{\Gamma(c-k+1)\Gamma(k+1)\Gamma(1-d)\Gamma(\alpha+d)\Gamma(c+\alpha)}.$$

Like in the Dirichlet process case, the full posterior count $\eta_v$ conditioned on all $N$ hash data structures is proportional to the product of the individual hash posteriors. Note that when the discount $d = 0$, and we recover the 2-parameter IBP [16], the query is very similar to that of the CRP case, except with the extra mass parameter $\gamma$. We can infer the parameters $\alpha, \gamma, d$ by maximizing the marginal likelihood [22].

## 5  Experiments

We now examine the Bayesian posterior query and point estimates obtained using the CM sketch applied to several data streams. In Appendix G, we present synthetic data results on data generated from a Dirichlet process random measure and a stable beta process random measure, and we also provide comparisons to a few count-min sketch extensions. In all experiments, we use a 2-universal hash family. We constructed a stream of tokens using the 20 Newsgroups data set, where the sketch was updated using the training data set ($M = 1467345$), and evaluated queries on the set of unique tokens in the test set, which had $53975$ elements. For each task, we examined several hash parameter settings of $N = 4, 5, 6$, with $J = 8000, 10000, 12000$. For the posterior distribution of the count, we used the Dirichlet process sketching model, inferring $\alpha$ via empirical Bayes.

**Posterior query examples.**  In Figure 3, we present a few posterior distributions returned by querying a few low, medium, and high frequency tokens. In each plot, we show the posterior distribution computed from $N = 4$ hash functions and $J = 12000$, and the true count is denoted alongside the posterior mean, median, and MAP (CM). In these examples, the MAP (CM) performs poorly for the low frequency example, and the shrinkage estimators (mean and median) provide better estimates than the MAP (CM) estimator. In the medium frequency example, the mean underestimates significantly, but the median, which provides a more modest amount of shrinkage, is a better estimator than the MAP (CM). In the high frequency example, the MAP (CM) estimator performs well, and so the shrinkage estimators underestimate the true count. More examples of posterior queries are available in the appendix.

**Point estimation results.**  Though our primary goal was to compute a posterior distribution, rather than a point estimator, we wanted to understand the behavior of posterior point estimates relative to the frequency of the tokens. For each estimator, we the measured the *relative error*, defined by $\frac{|\hat{\eta}_v - \eta_v|}{\eta_v}$ by the true counts $\eta_v$ for all tokens in the test set. In Figure 4, we plot the mean relative error, which is the average over the relative errors for each true count. Here we plot the mean relative error against the true count to get a sense of the performance of the various estimators on low and

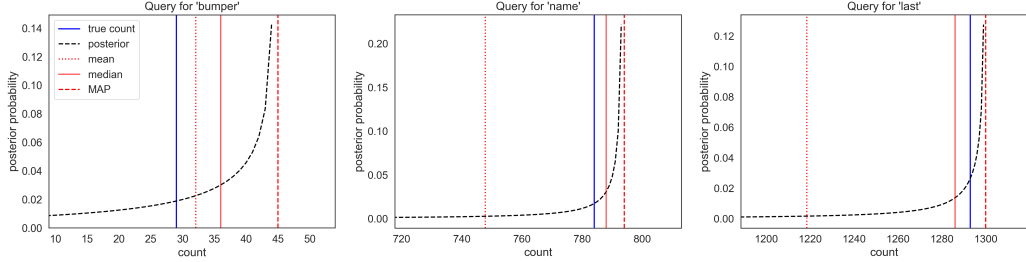

Figure 3: Posterior query for low, medium, and high frequency tokens The dashed black curve represents the posterior distribution returned by the query. The vertical blue line is the true count, and the vertical red counts denote the posterior mean, median, and MAP (CM estimator).

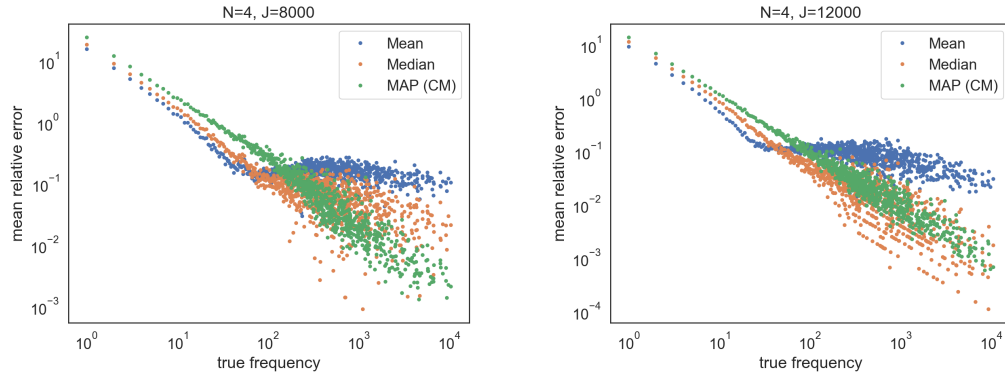

Figure 4: Mean relative error for point estimators of the posterior, constructed with hash parameters $J = 8000$ (left) and $J = 12000$ (right). Here we use $N = 4$ hash functions.

high frequency tokens. In this figure, we only show the $N = 4$ case, but additional results are in the appendix.

We can see that the mean is often a better estimator than the MAP (CM) and median for lower frequency events, especially when using less space. However, for medium frequency tokens, the posterior mean tends to underestimate the value of the count; instead, the posterior median provides a better estimate of the count. For very large frequencies, the MAP (CM) estimator generally performs well, as expected.

## 6 Conclusions and Future Work

We have introduced a Bayesian probabilistic view on the count-min sketch by taking the classical count-min sketch data structure and computing a posterior distribution over the counts. We show that, under the Dirichlet process, the MAP estimator recovers the count-min sketch estimator. We also demonstrate how similar Bayesian reasoning can be used to provide a posterior over the number of times a word appeared in a document generated from a beta-Bernoulli process. Many fruitful directions remain. On the theoretical side, extending this work to accommodate other token generating distributions for random partitions, such as the Pitman-Yor process, would be of interest for power law applications, as well as further exploration of the beta process sketch. Another direction of interest is using the posterior estimates of the counts and the linear sketch properties of the CM sketch for large-scale streaming algorithms, e.g., for large text or streaming graphs applications. Lastly, one may be interested in extending this method to accommodate different update operations, such as the conservative update, as well as different types of queries, such as range and inner product queries.

**Acknowledgments**

Michael Mitzenmacher was supported in part by NSF grants CCF-1563710, CCF-1535795, CCF-1320231, and CNS-1228598. Ryan Adams was supported in part by NSF IIS-1421780 and the Alfred P. Sloan Foundation.

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
