[Supplementary Material]

# Supplementary Material:
# A Bayesian Nonparametric View
# on Count-Min Sketch

**Diana Cai**
Princeton University
dcai@cs.princeton.edu

**Michael Mitzenmacher**
Harvard University
michaelm@eecs.harvard.edu

**Ryan P. Adams**
Princeton University
rpa@princeton.edu

## A Count-min sketch details

Recall Theorem 1 of Cormode and Muthukrishnan [2, Theorem 1] summarized in the main text: Let $J = \lceil \frac{e}{\epsilon} \rceil$ and $N = \lceil \log \frac{1}{\delta} \rceil$, with $\epsilon > 0, \delta > 0$. Then the estimate of the count $\hat{\eta}_x^{\mathrm{CM}}$ satisfies $\hat{\eta}_x^{\mathrm{CM}} \geq \eta_x$ and with probability at least $1 - \delta$, the estimate satisfies $\hat{\eta}_x^{\mathrm{CM}} \leq \eta_x + \epsilon M$.

*Proof.* Let

$$I_{n,k,l} := \begin{cases} 1 & \text{if } h_n(k) = h_n(l), \quad k \neq l \\ 0 & \text{otherwise} \end{cases}. \tag{A.1}$$

The expected value of $I_{n,k,l}$ is

$$\mathbb{E}(I_{n,k,l}) = \Pr(h_n(k) = h_n(l)) = \Pr(\bigcup_{j=1}^{J} \{h_n(k) = j, h_n(l) = j\})$$

$$\leq \sum_{j=1}^{J} \Pr(h_n(k) = j, h_n(l) = j) = \frac{1}{J},$$

due to a union bound and pairwise independence.

Now define

$$X_{kn} := \sum_{l=1}^{K} \eta_l \, I_{n,k,l}, \tag{A.2}$$

which is the number of times the $k$th item collides with some other type $l$ for the $n$th hash function, multiplied by the count $\eta_l$, which has expectation

$$\mathbb{E}(X_{kn}) = \mathbb{E}\left(\sum_{l=1}^{K} \eta_l \, I_{n,k,l}\right) = \sum_{l=1}^{K} \eta_l \, \mathbb{E}(I_{n,k,l}) \leq \frac{1}{J} \sum_{l=1}^{K} \eta_l = \frac{1}{J} ||\eta||_1. \tag{A.3}$$

Here $X_{kn}$ represents the noisy part of the the counts in $C$ that arises due to collisions. That is,

$$c_{n,h_n(k)} = \eta_k + X_{kn}.$$

This implies that $c_{n,h_n(k)} \geq \eta_k$ for all $n \in [N]$, and so each count cell in $C$ overestimates the value of the true count $\eta_k$. To see this, we can rearrange the probability of the event

$$\Pr(\hat{\eta}_k > \eta_k + \epsilon ||\eta||_1) = \Pr(\forall n \in [N], \{c_{n,h_n(k)} > \eta_k + \epsilon ||\eta||_1\})$$

$$\leq \Pr(\forall n \in [N], \{X_{kn} > \epsilon J \cdot \mathbb{E}(X_{kn})\}).$$

where the last inequality is from substituting Equation (A.3). Now by Markov's inequality, and letting $J = \lceil \frac{e}{\epsilon} \rceil$, we have

$$\Pr(\forall n \in [N], \{X_{kn} > \epsilon J \cdot \mathbb{E}(X_{kn})\}) = \Pr(\forall n \in [N], \{X_{kn} > e\mathbb{E}(X_{kn})\}) \leq \frac{1}{e^N}.$$

Choosing $N = \log(\lceil \frac{1}{\delta} \rceil)$, we have that $\frac{1}{e^N} \leq \delta$. Thus, the probability the estimate $\hat{\eta}_k$ is within an $\epsilon||\eta||_1$ additive factor of the true value $\eta_k$ is at least $1 - \delta$. $\qquad\square$

## B    Infinite multinomial occupancy scheme

In the infinite multinomial occupancy scheme (see Gnedin et al. [3]), we have frequencies $(p_j)_{j=1}^{\infty}$, where $\sum_{j=1}^{\infty} p_j = 1$, and we throw $n$ balls into the boxes, where each ball lands with probability $p_j$. Furthermore, we assume that the frequencies $(p_j)$ are random, drawn from a subordinator with independent increments with mean measure $\nu$. (Jumps by $p_j$ at rate 1 for each $j$.)

**Proposition B.1** (Gnedin et al. [3]). *The expected number of bins with $j$ balls in a sample of size $n$ is*

$$\mathbb{E}(\pi_{n,j}) = \binom{n}{j} \int_0^1 x^j (1-x)^{n-j} \nu(dx), \tag{B.1}$$

*where the rate measure $\nu$ is defined by $\nu(A) = \mathbb{E}(\sum_i \delta_{p_i}(A))$, and can be interpreted as the intensity measure of a subordinator, where we have jumps at $p_j$ with rate 1.*

*Proof.* Let $\pi_{n,j}$ be the number of bins with exactly $j$ balls: if $N_1, N_2 \ldots$ are the associated counts obtained from the $n$ draws,

$$\pi_j = \sum_{i=1}^{\infty} \mathbb{1}(N_i = j).$$

Then, its expectation is given by

$$\mathbb{E}(\pi_j) = \mathbb{E}\left(\binom{n}{j} \sum_{i=1}^{\infty} p_i^j (1-p_i)^{n-j}\right) = \binom{n}{j} \int_0^1 x^j (1-x)^{n-j} \nu(dx), \tag{B.2}$$

where $\nu(dx)$ is the mean measure, and the equality follows from

$$\mathbb{E}(\sum_i f(p_i)) = \int_0^1 f(x)\nu(dx)$$

holds for arbitrary $f \geq 0$. $\qquad\square$

Note that this expectation can also be applied to the case where $\sum_i p_i < \infty$ is not necessarily 1. See Gnedin et al. [3] for details, and Broderick et al. [1] for details on applying this to the stable beta process.

## C    Poisson processes

Kingman [4] provides an overview of properties of Poisson point processes. We outline a few useful results below. A fundamental property of Poisson processes is that the superposition of independent Poisson processes is itself a Poisson process. This is summarized by the theorem below.

**Theorem C.1** (Kingman [4]). *Let $\Pi_1, \ldots, \Pi_J$ be independent Poisson processes on $S$, and let $\Pi_j$ have mean measure $\mu_j$. Then the point process defined by their superposition*

$$\Pi = \bigcup_{j=1}^{J} \Pi_j$$

*is a Poisson process with mean measure $\mu = \sum_{j=1}^{J} \mu_j$.*

Below we state the coloring theorem.

**Theorem C.2** (Kingman [4]). *Let $\Pi$ be a Poisson process on $\Omega$ with mean $\mu$. Let the points of $\Pi$ be colored randomly with $J$ colors, and let the probability of a point receiving the jth color be $p_j$. Furthermore, suppose the colors of different points are independent. Let $\Pi_j$ be the set of points with the jth color. Then, the $\Pi_j$ are independent Poisson processes with mean measures $\mu_j = p_j\mu$.*

Note that the superposition of the $\Pi_j$ gets back $\Pi = \cup_{j=1}^J \Pi_j$ with mean measure $\sum_{j=1}^J \mu_j$.

# D   Proofs for Bayesian CM-sketch

**Lemma D.1** (Main text, Lemma 1). *Let $\pi$ be a $\mathrm{CRP}(\alpha)$-distributed random partition of $c$ items and $\pi_k$ be the number of subsets in that partition of size $k$. The expected number of size-$k$ subsets is*

$$\bar{\pi}_k := \mathbb{E}[\pi_k] = \frac{\alpha}{k} \frac{\Gamma(c+\alpha-k)\Gamma(c+1)}{\Gamma(c+\alpha)\Gamma(c+1-k)}. \tag{D.1}$$

*where $\Gamma(\cdot)$ is the gamma function.*

*Proof.* The Dirichlet process with concentration $\alpha$ has rate measure $\nu(dp) = \alpha p^{-1}(1-p)^{\alpha-1}dp$ which can be obtained by computing $p^{-1}\nu_1(dp)$, where $\nu_1$ is the structural distribution [6, 3]. We can apply Proposition B.1 to get

$$\mathbb{E}[\pi_k] = \frac{\Gamma(c+1)}{\Gamma(c-k+1)\Gamma(k+1)} \int_0^1 p^k(1-p)^{c-k} \, \alpha'p^{-1}(1-p)^{\alpha-1}dp$$

$$= \alpha \frac{\Gamma(c+1)}{\Gamma(c-k+1)\underbrace{\Gamma(k+1)}_{=k\Gamma(k)}} \mathrm{B}(k,c-k+\alpha) \underbrace{\int_0^1 p^{k-1}(1-p)^{c-k+\alpha-1}\frac{1}{\mathrm{B}(k,c-k+\alpha)}dp}_{:=1}$$

$$= \alpha \frac{\Gamma(c+1)\Gamma(k)\Gamma(c-k+\alpha)}{\Gamma(c-k+1)k\Gamma(k)\Gamma(c+\alpha)} = \frac{\alpha}{k}\frac{\Gamma(c+1)\Gamma(c-k+\alpha)}{\Gamma(c-k+1)\Gamma(c+\alpha)}.$$

$\square$

**Theorem D.1** (Main text, Theorem 2). *Let there be $N$ hash functions $h_1,\ldots,h_N$ each drawn uniformly at random from a truly random hash family $\mathcal{H}$. Define a Dirichlet process stream of tokens as in Proposition 1. When the Mth item $x_M$ arrives, let its nth hashed value be $h_n(x_M) = j_n$. Define $c_{n,j_n}$ to be the number of previous items in the stream that the nth hash has also hashed to $j_n$, i.e., $c_{n,j_n} = \sum_{m=1}^{M-1} \mathbb{1}[h_n(x_m) = j_n]$. Given the counts $c_{1,j_1},\ldots,c_{N,j_N}$, the posterior distribution over the true number of previous occurrences of items with the same type as $x_M$ is*

$$\Pr(\eta_x = k \,|\, c_{1,j_1},\ldots,c_{N,j_N},\alpha) \propto \prod_{n=1}^N \frac{1}{\alpha/J + c_{n,j_n}} \begin{cases} k\,\bar{\pi}_{k,c_{n,j_n}} & \text{if } k > 0 \\ \alpha/J & \text{if } k = 0, \end{cases} \tag{D.2}$$

*where*

$$\bar{\pi}_{k,c_{n,j_n}} = \frac{\alpha}{Jk}\frac{\Gamma(c_{n,j_n}+1)\Gamma(c_{n,j_n}-k+\alpha/J)}{\Gamma(c_{n,j_n}-k+1)\Gamma(c_{n,j_n}+\alpha/J)}. \tag{D.3}$$

*Proof.* Because of the assumption of independence of the hash family, we can factor the joint distribution of the observed counts into the product of the likelihoods of the individual hash function

count $c_n$:

$$\Pr(\eta_x = k | c_{1,j_1}, \ldots, c_{N,j_N}, \alpha') = [\Pr(c_{1,j_1}, \ldots, c_{N,j_N})]^{-1} \Pr(\eta_x = k) \prod_{n=1}^{N} \Pr(c_{n,j_n} | \eta_x = k, \alpha')$$

$$= [\Pr(c_{1,j_1}, \ldots, c_{N,j_N})]^{-1} \Pr(\eta_x = k) \prod_{n=1}^{N} \frac{\Pr(c_{n,j_n}, \eta_x = k | \alpha')}{\Pr(\eta_x = k)}$$

$$= [\Pr(c_{1,j_1}, \ldots, c_{N,j_N}) \Pr(\eta_x = k)^{N-1}]^{-1} \prod_{n=1}^{N} \Pr(c_{n,j_n}) \Pr(\eta_x = k | \alpha')$$

$$= [\Pr(\eta_x = k)^{N-1}]^{-1} \prod_{n=1}^{N} \Pr(\eta_x = k | \alpha')$$

$$\propto \prod_{n=1}^{N} \frac{1}{\alpha' + c_{n,j_n}} \begin{cases} k\, \bar{\pi}_{k,c_{n,j_n}} & \text{if } k > 0 \\ \alpha' & \text{if } k = 0, \end{cases}$$

where in the last line, we substitute the result from Proposition 1 and observe that $\Pr(\eta_x = k)$ is a small positive constant under the DP prior. $\qquad \square$

## E   Point estimates

**Lemma E.1** (Main text, Lemma 2)**.** *For $\alpha < J$, the function $p(\eta_x = k | c_{n,j_n}, \alpha')$ is strictly increasing on $k = 0, \ldots, c_{n,j_n}$.*

*Proof.* By our assumption that $\alpha < J$, we have that $\alpha' < 1$. For $k = 0$, the probability is proportional to $\alpha' < 1$. For $k = 1$, the probability is proportional to

$$\pi_{1,c_n} = \alpha' \frac{\Gamma(c_n + \alpha' - 1)\Gamma(c_n + 1)}{\Gamma(c_n + \alpha')\Gamma(c_n)} = \alpha' \frac{\Gamma(c_n + \alpha' - 1)(c_n)\Gamma(c_n)}{(c_n + \alpha' - 1)\Gamma(c_n + \alpha' - 1)\Gamma(c_n)} = \alpha' \frac{c_n}{c_n + \alpha' - 1} > \alpha',$$

where we used $\Gamma(x + 1) = x\Gamma(x)$, and the inequality follows from $\alpha' < 1$. Thus, the value of the function for $k = 1$ is greater than for $k = 0$.

Now the goal is to show for all $k = 2, \ldots, c_{n,j_n}, p(k | c_{n,j_n}, \alpha') > p(k - 1 | c_{n,j_n}, \alpha')$. We have that

$$\frac{k\, \pi_{k,n}}{(k-1)\pi_{k-1,n}} = \frac{k}{k-1} \frac{\alpha'}{k} \frac{\Gamma(c_{n,j_n} + \alpha' - k)\Gamma(c_{n,j_n} + 1)}{\Gamma(c_{n,j_n} + \alpha')\Gamma(c_{n,j_n} + 1 - k)} \frac{k-1}{\alpha'} \frac{\Gamma(c_{n,j_n} + \alpha')\Gamma(c_{n,j_n} + 1 - (k-1))}{\Gamma(c_{n,j_n} + \alpha' - (k-1))\Gamma(c_{n,j_n} + 1)}$$

$$= \frac{\Gamma(c_{n,j_n} + \alpha' - k)}{\Gamma(c_{n,j_n} + 1 - k)} \frac{\Gamma(c_{n,j_n} + 1 - k + 1))}{\Gamma(c_{n,j_n} + \alpha' - k + 1))}$$

$$= \frac{\Gamma(c_{n,j_n} + \alpha' - k)}{\Gamma(c_{n,j_n} + 1 - k)} \frac{(c_{n,j_n} + 1 - k)\Gamma(c_{n,j_n} + 1 - k))}{(c_{n,j_n} + \alpha' - k)\Gamma(c_{n,j_n} + \alpha' - k)} = \frac{(c_{n,j_n} + 1 - k)}{(c_{n,j_n} + \alpha' - k)} > 1,$$

since $\alpha' < 1$. Thus, $p(k | c_n)$ is (strictly) monotonically increasing on $k = 0, \ldots, c_{n,j_n}$. $\qquad \square$

From the last line of the proof, we can also see that the posterior is strictly decreasing when $\alpha > J$.

Using this result, we can describe the behavior of the posterior mean, median, and mode (MAP).

**Proposition E.1.** *For $\alpha < J$, we the following relationships between estimators:*

$$\hat{\eta}_x^{\mathrm{MAP}} = \hat{\eta}_x^{\mathrm{CM}}, \qquad \text{and} \qquad \hat{\eta}_x^{\mathrm{mean}} \leq \hat{\eta}_x^{\mathrm{med}} \leq \hat{\eta}_x^{\mathrm{MAP}}.$$

*Proof.* The MAP estimator is given by the $k$ that maximizes a product of monotone functions (from $k = 0, \ldots, c_{n,j_n}$), i.e.,

$$\hat{\eta}_x^{\mathrm{MAP}} := \arg\max_k p(\eta_x = k | c_{1,j_1}, \ldots, c_{N,j_N}, \alpha') = \arg\max_k \prod_{n=1}^{N} p(\eta_x = k | c_{n,j_n}, \alpha'),$$

where $j_n := h_n(x)$ is the bucket that the $n$th hash sends $x$ to.

By Lemma 2, the posterior of each hash $h_n$, $p(\eta_x = k | c_{n,j_n}, \alpha')$, is monotonically increasing in $k$, for $k = 0, \ldots, c_{n,j_n}$. Furthermore, each individual posterior is equal to 0 for $k > c_n$, since the true count $n_k$ cannot exceed the total colliding mass $c_n$.

Thus, this full posterior, which is proportional to the product of the individual hash functions' posteriors, is monotonically increasing in $k = 0, \ldots, \min_{n \in [N]} c_{n,j_n}$ since for each all hash functions $h_n$, $p(\eta_x = k | c_{n,j_n}, \alpha')$ is monotone on $k = 0, \ldots, \min_{n \in [N]} c_{n,j_n}$. Since, for the hash function $h_{n'}$ with the minimum count $c_{n'}$, the posterior is 0 for any $k > c_{n',j_{n'}}$, the product of all of the hashes' posteriors must be 0 for $k > \min_{n \in [N]} c_{n,j_n}$.

Thus, we have that the MAP estimate is equal to the minimum count, i.e.,

$$\hat{\eta}_x^{\text{MAP}} = \min_{n \in [N]} c_{n,j_n} := \hat{\eta}_x^{\text{CM}},$$

which is the estimator given by the CM-sketch.

The relationship $\hat{\eta}_x^{\text{mean}} \leq \hat{\eta}_x^{\text{med}} \leq \hat{\eta}_x^{\text{MAP}}$ follows from the fact that the distribution is skewed, as it is monotonically increasing. $\qquad\square$

# F  Details for stable beta-Bernoulli process sketching

Recall the model given by stable beta-Bernoulli process (BP)

$$B \sim \text{BP}(\alpha, \gamma, d), \qquad X_m | B \overset{\text{iid}}{\sim} \text{BeP}(B), \tag{F.1}$$

where $\alpha > 0$ is the concentration parameter, $\gamma > 0$ is the mass parameter, $d \in [0, 1), d > -\alpha$ is the discount parameter. The condition $d > -\alpha$ ensures that the total mass is finite (and therefore each $X_m$ has a finite number of non-zero features). For additional details on the stable beta process (aka 3-parameter beta process), see Teh and Görür [7], Broderick et al. [1]. For additional details on beta processes, see Thibaux and Jordan [8].

The stable beta process is a completely random measure, which can be represented by a Poisson point process with rate measure

$$\nu(dw) = \gamma \frac{\Gamma(1 + \alpha)}{\Gamma(1 - d)\Gamma(\alpha + d)} w^{-1-d}(1 - w)^{\alpha + d - 1} dw,$$

which governs the distribution of the weights. A separate base measure can be specified, but for this paper, we assume the atoms $\omega$ are distributed uniform in $\Omega$.

As in the Dirichlet (gamma) process case, we can apply the Poisson coloring theorem to get $J$ independent beta-Bernoulli processes with mass parameter $\gamma' = \gamma/J$. Analogous to the DP case, this is used then used to model the sketch data generating process with a beta-Binomial process with $M$ draws.

To compute the posterior BP query, we need the predictive rule from the 3-parameter Indian buffet process (3-IBP), the marginal process of the stable beta process. In the IBP analogy, customers are observations, and dishes represent features. The 3-parameter Indian buffet process gives the probability of seeing an existing feature $i$ in the next observation as $\Pr(x_{n+1,i} = 1 | x_{1,i}, \ldots, x_{n,i}) = \frac{\eta_i - d}{n + \alpha}$, and then $\text{Pois}(\gamma \frac{\Gamma(1+\alpha)\Gamma(n+\alpha+d)}{\Gamma(n+1+\alpha)\Gamma(\alpha+d)})$ new features are drawn. Here $\eta_k$ is the number of times a customer has tried the $k$th dish.

Note that when the discount $d = 0$, we recover the 2-parameter IBP (2-IBP):

$$\Pr(x_{n+1,i} = 1 | x_{1,i}, \ldots, x_{n,i}) = \frac{\eta_i}{n + \alpha}$$

for an existing dish $i$ and then $\text{Pois}(\frac{\gamma\alpha}{n+\alpha})$ new dishes are drawn.

Now the conditional probability of the count of feature $x \in \mathbb{N}$ in a new observation being $k$ conditioned on the number of items that hashed into the same bucket $c_{h(x)}$ and the number of features in that bucket with count $k$ is:

$$p(\eta_x = k | \pi_{k,c_{h(x)}}, c_{h(x)}) = \begin{cases} \pi_{k,c_{h(x)}} \frac{k-d}{c_{h(x)}+\alpha} & \text{if } k > 0 \\ \gamma \frac{\Gamma(1+\alpha)\Gamma(c_{h(x)}+\alpha+d)}{\Gamma(c_{h(x)}+1+\alpha)\Gamma(\alpha+d)} & \text{if } k = 0 \end{cases},$$

Figure 1: Beta-Bernoulli process posterior for $\gamma' = 1$ and $c = 50$ for a single hash function. The dashed line denotes the posterior mean.

where $\pi_{k,n}$ is the number of tokens with count $k$ in a bin with $n$ tokens hashing to it.

Thus, we have the individual hash's posterior

$$p(\eta_x = k|c_{h(x)}) = \sum_{\pi_k} p(\eta_x = k|\pi_{k,c_{h(x)}}c_{h(x)})p(\pi_k|c_{h(x)}) = \begin{cases} \bar\pi_k \frac{k-d}{c_{h(x)}+\alpha} & \text{if } k > 0 \\ \gamma \frac{\Gamma(1+\alpha)\Gamma(c_{h(x)}+\alpha+d)}{\Gamma(c_{h(x)}+1+\alpha)\Gamma(\alpha+d)} & \text{if } k = 0 \end{cases},$$

where

$$\bar\pi_{k,n} \approx \binom{n}{k}\gamma\int_0^1 w^k(1-w)^{n-k}\gamma\frac{\Gamma(1+\alpha)}{\Gamma(1-d)\Gamma(\alpha+d)}w^{-1-d}(1-w)^{\alpha+d-1}dw$$

$$= \gamma\frac{\Gamma(n+1)\Gamma(1+\alpha)\Gamma(k-d)\Gamma(n-k+\alpha+d)}{\Gamma(n-k+1)\Gamma(k+1)\Gamma(1-d)\Gamma(\alpha+d)\Gamma(n+\alpha)}.$$

Note that when the discount $d = 0$, and we recover the 2-IBP, the query is very similar to that of the CRP case, except with the extra mass parameter $\gamma$. The posterior count conditioned on all $N$ counters is then proportional to the product of the individual hash posteriors.

In Figure 1, we plot the posterior for a single hash function. The vertical line denotes the posterior mean. Lighter colors (higher values of $d$) represent heavier-tailed distributions, and we can see that higher values of $d$ correspond to the posterior assigning more mass to smaller counts. For $d = 0, \alpha < 1$, we again observe the monotonicity of the posterior. We can also see that as the concentration parameter $\alpha$ grows, the posterior places more mass on lower counts; this reflects the distribution spreading the mass of the random measure across more atoms.

## G  Additional experiments and details

For all experiments, we used a 2-universal hash family, given by the construction below. All code was implemented in the Julia programming language (v0.6), and experiments were run on a laptop.

### G.1  Hash family construction

For all experiments, we used the following construction for a 2-universal hash family. Fix $K, J$. Let $p \geq K$ be a prime number. Let $g : \mathbb{Z}_p \to [J]$ be the function given by $g(x) = x \mod J$. For all $a, b \in \mathbb{Z}_p$, define the linear function $f_{a,b} : \mathbb{Z}_p \to Z_p$ as

$$f_{a,b}(x) = ax + b \,(\bmod\, p).$$

(a) Low frequency example        (b) High frequency example

Figure 2: Simulations for the Bayesian count-min sketch with CRP-distributed tokens for a low and high frequency token example. The dashed black curve is the full posterior over $N = 4$ hash functions. The vertical red lines denote the mean, median, and MAP estimators, and the blue vertical line is the true count. Here $\alpha$ is assumed to be known.

Finally, define the hash function $h_{a,b} : \mathbb{Z}_p \to [J]$ as $h_{a,b}(x) = g(f_{a,b}(x))$. The family of hash functions defined by $\mathcal{H} = \{h_{a,b} : a, b \in \mathbb{Z}_p \text{ with } a \neq 0\}$ is a *2-universal family* [5, Thm. 8.16], i.e., for all $x, y \in [K]$ such that $x \neq y$, and $h$ chosen uniformly at random from $\mathcal{H}$,

$$\Pr_{h \in \mathcal{H}} (h(x) = h(y)) \leq \frac{1}{J}.$$

This also can be applied to the restriction of $\mathbb{Z}_p$ to $[M]$. To generate a hash function $h_{a,b}$ from $\mathcal{H}$, we choose $a, b$ uniformly at random from $\mathbb{Z}_p$, and then we can easily compute $h_{a,b}$.

### G.2 Synthetic data

We generated several synthetic datasets in order to better understand the behavior of the posterior queries and our theoretical results.

#### G.2.1 Dirichlet process simulations

**Data generation**    We simulated a synthetic data from the Chinese restaurant process with concentration parameter $\alpha = 3000$. We generated $M = 5 \times 10^4$ tokens for constructing the sketch, and then conditioned on those tokens, another $M' = 1 \times 10^3$ tokens for query processing. The total number of unique tokens was 15,562.

**Sketch construction**    For a fixed $N, J$, we first generated $N$ hash functions from the 2-universal family described above, where the domain was set to the largest possible number of unique tokens, and chose a prime larger than that. For the first $M$ data points, we hashed each data point and incremented the associated counters (i.e., performed the update operation for the sketch).

**Posterior queries**    Figure 2 shows the posterior for a low and high frequency example, highlighting the posterior mean, median, and MAP. In these examples, we can see that posterior mean provides a better estimate than the MAP, as it the averaging takes into account uncertainty from the posterior probability mass placed on other counts.

We also looked at the query of the same token but under different sketches (each sketch with a different random seed) and how the posterior changes depending on the randomness of the hash functions drawn. For clarity of presentation, we show two runs per query.

**Predictive distribution approximation**    Under the Dirichlet process, the predictive distribution of a new token is

$$p(x_{M+1}|x_1, \ldots, x_M) = \frac{1}{M + \alpha} \begin{cases} \eta_{x_{M+1}} & \text{if } x_{M+1} \text{ is an existing token} \\ \alpha & \text{if } x_{M+1} \text{ is a novel token} \end{cases},$$

Figure 3: Posterior distributions generated under 2 different sketches both with $N = 4, J = 4000$, denoted by black and gray lines. The true count is indicated by the blue solid line. Here we show three example token queries for a low, medium, and high frequency token.

Figure 4: Predictive distribution approximation errors induced from using the full posterior and averaging out the count vs plug-in point estimators given by the posterior mean, median, and MAP (CM).

where $\eta_{x_{M+1}}$ is the number of times we observed that token in the first $M$ elements.

Now, under the sketch, we are throwing away the information in $x_1, \ldots, x_M$ and replacing it with the sketch $(c_{n,j})$, thereby introducing uncertainty in the value of the count $\eta_{x_{M+1}}$.

One way of approximating this predictive distribution is to approximate the sufficient statistic using a point estimator, such as the count-min sketch estimator or the posterior mean/median of our method. To do so, we replace $\eta_{x_{M+1}}$ with its estimate $\hat{\eta}_{x_{M+1}}$.

Alternatively, we can consider averaging over the value of the count $\eta$

$$\sum_{\eta_{x_{M+1}}} p(x_{M+1}, \eta_{x_{M+1}} | (c_{n,j})) = \sum_{\eta} p(\eta_{x_{M+1}} = \eta | (c_{n,j})) \, p(x_{M+1} | \eta_{x_{M+1}} = \eta),$$

where, on the right hand side, the first term is the posterior distribution over the count that is output from our method, and the second term is the predictive distribution evaluated at a particular value of $\eta_{x_{M+1}} = \eta$.

We compared the predictive distribution approximation errors from averaging over the posterior with the predictive distributions induced by the plug-in estimators from the posterior mean, median, and MAP. We computed the exact predictive distribution and then computed the relative error between the true predictive distribution and each approximation, where we computed one relative error for each token in the query "test" set.

**Point estimation results**   For the sketches that we constructed, varied the sketch width $N = 3, 4$ and depth $J = 4000, 4800, 5600$. In Figure 5, we plotted the relative error, defined by $\frac{|\hat{\eta}_v - \eta_v|}{\eta_v}$ by the true counts $\eta_v$ for all tokens in the test set. Here we plot the relative error against the true count to get a sense of the performance of the various estimators on low and high frequency tokens. Recall that the MAP is the same as the count-min sketch estimator.

(a) $N = 3, J = 4000$  (b) $N = 3, J = 4800$  (c) $N = 3, J = 5600$

(d) $N = 4, J = 4000$  (e) $N = 4, J = 4800$  (f) $N = 4, J = 5600$

Figure 5: Relative error measured by absolute difference between estimate and true count, divided by the true count, on a held out test set.

## G.3  Count-min sketch extensions

**Conservative updates**   We show results on the synthetic data of the posterior point estimates in comparison to the estimates obtained from count-min sketch and conservative updates, where counters are only updated when needed. That is, the update only increments minimum counter(s). Conservative updates worked well on our example for high-frequency counts, far exceeding the CM-sketch estimator (and therefore the MAP estimator). We display some of the low frequency count results, showing that the posterior mean has comparable or slightly worse relative error than conservative updates. In future work, computing a posterior distribution over the counts from a sketch constructed using conservative updates is desirable to quantify the uncertainty in the count.

**Count-mean-min**   In the conservative updates setting, the update rule is changed but the point query is the same. By contrast, like in our Bayesian CM setting, the count-mean-min estimator assumes the same update rule as CM but has a different point query. The aim of the count-mean-min estimator is to subtract the an estimate of the bias from the count-min estimator. We compared to the count-mean-min estimator and found that it tended to underestimate the count, even more so than the posterior mean, for the same settings of hash parameters.

### G.3.1  Beta process simulations

For the beta-Bernoulli process, we generated data from a 3-parameter Indian buffet process with parameters $\gamma = 400, \alpha = 0.5, M = 2000$. We generated one dataset with $d = 0$ (no power law, left figure) and one with $d = 0.2$ (slight power law, right figure). We fit the sketch with $N = 3, J = \gamma$. In Figure 7, we plot the posterior along with the posterior mean and the count given by count-min sketch. As we can see from the true count, the estimate from CM-sketch is significantly overestimating the true count.

### G.4  Text data

As stated in the main paper, we constructed a stream of tokens using the 20 Newsgroups data set, where the sketch was updated using the training data set ($M = 1{,}467{,}345$), and evaluated queries on the set of unique tokens in the test set, which had $53{,}975$ elements. For each task, we examined several hash parameter settings of $N = 4, 5, 6$, with $J = 8000, 10000, 12000$. For the posterior distribution of the count, we used the Dirichlet process sketching model, inferring $\alpha$ via empirical Bayes.

Figure 6: Low frequency counts generated from the $\mathrm{CRP}(\alpha)$. The posterior point estimates are compared to the point estimator from the count-min sketch with conservative updates.

Figure 7: Posterior over count of tokens from data generated from a beta-Bernoulli processes, where $d = 0, 0.2$, respectively.

In Figure 8, we provide a few additional visualizations of various posterior queries for tokens, representing low, medium, and high frequency token examples. Here we see that the posterior mean is a useful estimator for low frequency counts, the median may provide a better estimate than the MAP (CM) for medium and high frequency counts. In the query for religion, we see that the MAP performs quite well already, and thus, the mean and medium will underestimate, due to the monotonicity property of the posterior.

**Additional point estimation results**   In Figure 9, we plot additional point estimation results for the other hash parameter settings. We see the same trends summarized in the main document: the posterior mean provides the most shrinkage and therefore tends to perform well on low frequency counts, whereas medium and high frequency counts often benefit from the more modest shrinkage posterior median, as the CM sketch tends to perform well on large counts. Additionally, we see that the posterior mean tends to be useful (relative to the MAP) in the cases when we use less space $J = 8000$.

Figure 8: Additional posterior queries for low, medium, and high frequency tokens.

Figure 9: Mean relative error for point estimators of the posterior, constructed with $N$ hash functions with a range of size $J$.

**Estimation of** $\alpha$    In Figure 10, we plot the marginal likelihood (Equation (10)) and the inferred value of $\alpha$ for the $N = 4, J = 12000$ hash setting. We computed the maximum marginal likelihood using the optimize function in the Julia package `Optim.jl`.

**Query time**    We measured the query times to get a sense of the additional time cost computing the posterior distribution has over the count-min sketch point query. Here we considered the settings of $N = 5, J = 10000$. We constructed the sketch using the entire training data set and then queried the counts for the test set. Averaged over all timings in the test set, the average query time for the

Figure 10: Marginal likelihood of $\alpha$ for the $N = 4, J = 12000$ sketch.

count-min estimator was $7.5e - 6$ seconds, and the average query time for the posterior distribution was $2.5e - 4$ seconds. While this is much slower, the latter is still computationally efficient, in practice. Note that the timings are based on an implementation in Julia on a standard laptop.