[Reviews · NeurIPS 2018]

Reviewer 1



Summary A Bayesian nonparametric approach is derived to find the posterior distribution on the number of tokens of type k observed using the information in a count-min sketch. A Dirichlet process is used as a prior on the relative frequency of observing different tokens and the posterior distribution is derived. The posterior has an analytic form and the authors show that the MAP estimator is the count-min estimate. The effect of the concentration parameter on the estimate is discussed. The idea is then applied to token within documents using a Beta-Bernoulli construction. The effect of using the posterior mean, rather than the MAP estimator, are illustrated on some simulated data sets. Strengths The idea of thinking about sketches in terms of Bayesian nonparametric is interesting to me. The combination of an infinite-dimensional process with suitable independence structure (the underlying Poisson random measure) is promising and has the potential to give analytic posterior (which is key for their use in these problems). The authors derive a nice set of results which give a clear idea of how the method works. Weaknesses The paper is fairly clearly written but I found it difficult to understand the count-min sketch from the authors' description. To me the main weakness of the paper is that it's unclear where this work leads. It would have been interesting to see how the method would work on more complicated queries than the number of time that a particular token has been observed. The paper also doesn't illustrate what's gained by replacing a point estimate by a posterior distribution.

Reviewer 2



The count-min (CM) sketch uses a set of hash functions to efficiently estimate frequencies of large item sets. The primary contribution of this paper is to establish a clever and interesting connection between the CM frequency estimates, and the posterior of a Bayesian nonparametric prior. The biggest practical advantage of this conceptual link is to provide uncertainty estimates for the sketched counts (due to hash collisions, they are approximate). STRENGTHS: TECHNICAL CONTRIBUTIONS Some interesting technical work (which looks correct, and exploits independence properties of underlying Poisson processes) is needed to compute the posterior on true frequencies given a Dirichlet process (DP) prior and sketch-based likelihood. An extension to the stable beta-Bernoulli process allows sketching of multi-token documents and power-law priors, and hyperparameters may be estimated via empirical Bayes. For natural hyperparameters, the mode of the posterior corresponds to the classic CM-sketch estimate, while the mean or median are smaller (to account for the likely fact that some hashed counts are collisions) and often more accurate. Writing is dense but generally clear, with some important technical details supported by appendices. WEAKNESSES: EXPERIMENTS There are several experiments in the main paper, and more in plots in the appendices. While the simplest of these help to build intuition, in general they are a bit disorganized and hard to follow, and there is a shortage of explanation and discussion. More work is needed to guide the reader through these plots and draw out important insights (the paper ends abruptly with no conclusion). It was disappointing that almost all experiments use simple synthetic datasets, generated from the assumed DP (or beta-Bernoulli) priors. The final figure in the appendix uses real text data, but the experiment is only described by 3 high-level sentences that make some details unclear (e.g., it seems like the DP model was used, but the beta-Bernoulli would be more appropriate for the power law statistics of text documents). The final figure suggests that the mean estimator is less effective on this real data than it was on some synthetic data, but the median estimator still seems to provide a solid improvement on the CM baseline. I would have rated this paper as a strong accept if it had a more solid set of experiments, well integrated and discussed in the main text. FEEDBACK: Thank you for the clear and helpful author feedback. The manuscript would be much better with an improved presentation and discussion of the experiments, please do make the suggested changes in future revisions.

Reviewer 3



Very good write up and presentation. Theorem 2 is a very nice result, and an ingenious application of DP theory, demonstrating a good knowledge of relevant theory. No further comments here. Except, your approach requires some overhead for every token whose probability is estimated, a lot more than simply the min() function of current approach. So this should be discussed and demonstrated to show it is reasonable (which I expect it to be). But the simulations where disappointing. You simulate data according to known theory and of course the results match expectations. Why bother? What you should do here is embed your estimations into a real live run of CM-sketch and compare the results, also giving the additional compute time. The poor experimental work, which should be easily fixed, makes the paper "rejectable". References: "nlp", "dirichlet"